# Research on Environmental Governance, Local Government Competition, and Agricultural Carbon Emissions under the Goal of Carbon Peak

**Yingya Yang [1], Yun Tian [2,*], Xuhui Peng [3], Minhao Yin [2], Wei Wang [4] and Haiwen Yang [1]**

[1]   Business School, Anyang Institute of Technology, West Section of Huanghe Avenue, Anyang 455000, China
[2]   School of Business Administration, Zhongnan University of Economics and Law, No. 182, Nanhu Avenue, Wuhan 430073, China
[3]   Party School of the Wuxi Municipal Committee of CPC, 1 Yuanzhu Rd., Wuxi 214086, China
[4]   College of Management, Sichuan Agricultural University, 211 Huimin Rd., Chengdu 130062, China
*   Correspondence: tianyun1986@zuel.edu.cn; Tel.: +86-027-88386757

**Abstract:** By introducing network game theory, this paper analyzes the internal relationship and interaction mechanism among environmental governance, local government competition, and agricultural carbon peak level. On the basis of theoretical analysis, a spatial panel model is constructed using panel data from 30 provincial-level regions in China for empirical analysis. The research finds that local governments have positive competition with respect to the agricultural carbon peak, they adopt complementary carbon peak competition strategies, and they are more inclined to take geographical distance to adjacent regions as the yardstick in the competition with respect to the agricultural carbon peak strategy. That is, when the carbon peak level of surrounding provinces increases, the carbon peak level of the region will also increase. Thus, there is a phenomenon of mutual imitation and convergence between neighboring provinces. Environmental governance has a significant positive direct effect and a positive spatial spillover effect. From the perspective of coefficient, its direct effect is significantly greater than the spatial spillover effect. Therefore, more attention should be paid to local environmental governance to promote the improvement of the agricultural carbon peak level. Furthermore, the agricultural industrial structure, fiscal decentralization, agricultural public investment, regional industrial structure, and the proportion of the rural population have significant spatial spillover effects. The agricultural industrial structure and fiscal decentralization are significantly positively correlated with the peak level of agricultural carbon while the proportion of the rural population is significantly negatively correlated with the peak level of agricultural carbon. The research results have important theoretical value for expanding the research in the field of agricultural carbon emissions and provide important practical reference for China to successfully achieve the goal of agricultural carbon peak and promote the high-quality development of agriculture

**Keywords:** local government competition; environmental governance; agricultural carbon peak; network game model; spatial Dubin panel model





## 1. Introduction

In recent years, the climate change caused by greenhouse gas emissions has caused a series of impacts on the natural ecosystem and human living environment [1], which has gradually attracted extensive attention from all over the world [2]. After China acceded to the United Nations Framework Convention on climate change with the approval of the National People's Congress in November 1992, following the initial passive response, China submitted to the United Nations General Assembly in 2015 its independent emission reduction commitment to reduce its carbon emission intensity by 60–65% by 2030 compared with that in 2005. By 2020, China proposed the initiative to put forward carbon peak and carbon neutral targets at the United Nations, so as to actively fulfill its emission reduction

obligations as a developing country. China's carbon emission intensity has continued to decline since the 12th Five-Year Plan period. Compared with 2005, it decreased by nearly 50% in 2020. Good progress has been made in reducing carbon emissions. However, according to the report of the International Energy Agency, China's carbon emission intensity in 2021 is still at a high level compared with developed countries, and there is still room for further effort in the implementation of the "double carbon" goal.

As a typical public resource, environmental protection cannot be separated from the government's constraints and control [3]. The government has an unshirkable obligation to deal with environmental problems including carbon emissions. China's carbon emission intensity has been attached great importance by the central government, as well as governments at all levels, since its incorporation into the national economic and social development plan as a binding indicator during the 12th Five Year Plan. The traditional performance assessment mechanism is gradually changing, and the assessment of energy consumption, carbon emissions, and other indicators is gradually being strengthened. Since 2021, China has established the strategic direction of ecological civilization construction by focusing on carbon reduction, with the establishment of a special leading group for carbon peak and carbon neutralization. In the future, all national policy tools and organizational arrangements will focus on carbon reduction and emission reduction. At the same time, local governments have also responded to the efforts of the central government. Provinces (municipalities directly under the central government and autonomous regions) have successively set up carbon peak and carbon-neutral leading groups. In addition, some local governments have also added carbon emission intensity indicators to relevant assessment indicators. For example, Shandong Province includes carbon emission intensity indicators in the comprehensive performance assessment of high-quality development of various cities while Jing'an District of Shanghai has formulated work assessment measures related to the 2022 "double carbon" target. The competitive behavior of local governments will have an important impact on the intensity of carbon emissions [4]. Scholars have gradually paid increasing attention to the impact of the competitive behavior of local governments with respect to carbon emissions.

The relevant theories on local government competition can be traced back to Adam Smith's period. Since then, many scholars have conducted corresponding research on the competitive behavior between governments from the perspective of public goods such as taxation based on the public choice theory [3,5–10]. Since the reform of the tax-sharing system in 1994, China has gradually formed a unique Chinese-style power-sharing system that combines political centralization and economic decentralization. Under this system, local governments directly or indirectly control a large amount of economic resources and play an increasingly important role in regional economic development and social governance. This institutional arrangement gives local governments greater rights to occupy and control economic resources, in addition to bearing most of the expenditure responsibilities. The responsibility for environmental governance falls more on local governments [11]. In 2020, the State Council issued a reform plan on the division of central and local fiscal authority and expenditure responsibility in the field of ecological environment, which specifies the control of greenhouse gas emissions and other matters within local administrative areas, recognizing them as the local fiscal authority, with the local government bearing the expenditure responsibility.

As the main body of environmental governance, local governments have more common strategic interaction in the formulation and implementation of environmental policies [12–17]. According to the existing research, there are two kinds of competition incentive mechanisms for environmental governance among local governments: bottom-up [18] and top-down [19]. Specific to the reality of China, strategic interaction between local governments mainly comes from political incentives, with performance evaluation playing a pivotal role. Performance assessment including environmental protection indicators has a positive impact on the promotion incentive of local officials [20]. Under the Chinese-style decentralization, performance assessment directly affects the implementation effect of envi-

ronmental policies [21]. Some scholars believe that local governments relax environmental regulation standards for the sake of economic benefits and other considerations, which will lead to a decline in local environmental quality, whereas competition among governments will increase local pollution emissions [22–24], aggravate environmental pollution [25], and lead to the phenomenon of a "race to the bottom". Zhang Z. has found through research that the competition situation of environmental governance among local governments is changing to strategic imitation, and this competition strategy is closely related to the change in official performance assessment indicators [26]. Strict environmental performance indicators will encourage local governments to imitate each other [27], having a certain enhancement effect on the "competitive upward" strategic behavior among cities [28].

Because of its own characteristics, agriculture is not only a huge carbon sink system but also one of the main emission sources of greenhouse gases [29,30]. The greenhouse gases emitted from agricultural production activities are mainly methane and nitrous oxide. The emissions of these two gases from agricultural production activities account for a high proportion of the total emissions. According to the content of the second National Information Circular of the People's Republic of China on climate change, in 2005, methane emissions from agricultural production activities accounted for 56.62% of China's total emissions, and nitrous oxide emissions accounted for 73.79% of China's total emissions [31]. The heat absorption efficiency of these two gases is higher than that of carbon dioxide, and their influence on promoting global temperature rise is more obvious. In addition to paying attention to the carbon emissions of the secondary and tertiary industries, the issue of agricultural carbon emissions is also an important issue that China must solve to achieve the "double carbon" goal. In the government work report of the Central People's Government of the People's Republic of China in 2022, it is emphasized to further promote the reduction in and efficiency of chemical fertilizers and pesticides, as well as the resource utilization of livestock and poultry breeding wastes, to develop low-carbon agriculture, and to promote agricultural carbon reduction and emission reduction [32]. There are many existing studies on agricultural carbon emissions and their reduction. The relevant studies mainly focused on the construction of agricultural carbon emission index systems [33,34], the calculation of agricultural carbon emissions [35–37], the efficiency of agricultural carbon emissions [38–40], and the influencing factors of agricultural carbon emissions [41–43]. According to the relevant literature, it can be found that the existing research on the competitive behavior of local governments in environmental aspects focused on environmental pollution control and environmental regulation while the research on environmental governance and carbon emissions also focused on carbon emissions in the industrial field. The existing research on the competitive behavior of local governments in agricultural carbon emission reduction is relatively scarce, with little direct elaboration on the specific mechanism of local government competition in the pursuit of the agricultural carbon peak.

This paper mainly studies the driving mechanism of the local government to achieve the agricultural carbon peak goal, the impact of environmental governance on the agricultural carbon peak level, and the impact mechanism of local government competition. Through the introduction of network game theory, this paper analyzes the strategic interaction mechanism of environmental governance, local government competition, and agricultural carbon peak goal. On the basis of theoretical analysis, the panel data of 30 provincial-level regions in the Chinese Mainland (excluding Hong Kong, Macao, and Taiwan) are used to construct a spatial panel model for empirical analysis. The research results can provide a reference for effectively promoting the implementation of carbon reduction and emission reduction in China's agricultural sector, successfully achieving the carbon peak goal, and ultimately mitigating global warming.

## 2. Environmental Governance, Local Government Competition, and Agricultural Carbon Peak: A Network Game Model

As the decision maker, the local government's behavior is affected by the behavior of neighboring local governments in the game process. On the basis of the research of Peng X. et al. [44], this paper introduces the network game model to analyze the competitive behavior of local governments in achieving the goal of agricultural carbon peak.

Suppose there are $N$ local governments in a certain geographical area, $N = \{1, \cdots, n\}$, $n \geq 2$. If $N$ local governments are abstracted into $N$ nodes in the social network, a social network $g$ in the geographical space can be formed. In this network, there are a series of interconnected relationships among $N$ local governments, which can be expressed by the adjacency matrix $G = [g_{ij}]$. If $i$ is adjacent to $j$, then $g_{ij} = 1$; otherwise, $g_{ij} = 0$. Further, the network is an undirected network, i.e., $g_{ij} = g_{ji}$. In general, it is assumed that $g_{ii} = 0$, i.e., the local government i is not adjacent to itself, as reflected on the adjacency matrix, whereby the diagonal elements of $G$ are all 0. In this network, the number of neighbors of local government $i$ is expressed by $d_i$, which is the degree of the network node $i$.

This paper assumes that the agricultural carbon peak level of each local government is a continuous decision variable $c_i$. When $c_i \geq 0$, the spatial relationship network formed between local governments is represented by $G$. Then, the benefit function of each local government to achieve the agricultural carbon peak goal can be expressed as

$$u_i(c, g) = [\sigma \alpha_i + \zeta_i(X)]c_i - \frac{1}{2}c_i^2 + \psi \sum_{j=1}^{n} g_{ij}c_i c_j, \tag{1}$$

where $\sigma > 0$, and $\psi > 0$. $\psi$ reflects the strategic interaction of local governments in the agricultural carbon peak target action. Combined with the actual content of this study, it is assumed that local governments have strategic complementary behavior in the agricultural carbon peak target action. $\alpha_i$ is the unobservable heterogeneity of local governments, and $\zeta_i(X)$ is the heterogeneity of the captured exogenous decision. For the sake of simplifying the model, this paper only introduces the heterogeneity-influencing factor of environmental governance, which is an exogenous decision; thus, the specific expression of $\zeta_i(X)$ can be written as

$$\zeta_i(X) = \beta_0 Env_i + \frac{1}{d_i(g)}\gamma_0 \sum_{j}^{n} g_{ij} Env_j, \tag{2}$$

where $Env_j$ is the environmental governance variable that determines the observable influence factor of the heterogeneity of the agricultural carbon peak target income of each local government, $\beta_0$ and $\beta_1$ are relevant parameters, and $d_i$ is the degree of the node of local government $i$, i.e., the number of neighbors in the network with direct neighbor relations.

This paper uses Katz–Bonacich centrality to examine the importance of nodes, by defining

$$M = (E - \psi G)^{-1} = \sum_{k=0}^{+\infty} \psi^k G^k. \tag{3}$$

Then, the Katz–Bonacich centrality of individual $i$ is defined as follows:

$$b_i(g, \psi) = \sum_{j=1}^{n} g_{ij} = \sum_{j=1}^{n} \sum_{k=0}^{+\infty} \psi^k g_{ij}^{[k]}. \tag{4}$$

For simplicity, it can be expressed in matrix form as follows:

$$b(g, \psi) = M\mathbf{1} = (E - \psi G)^{-1}\mathbf{1}, \tag{5}$$

where $\mathbf{1}$ is the n-dimensional unit vector, and $E$ is the unit matrix. Similarly, the centrality of the weighted Katz–Bonacich network can be obtained as follows:

$$b_\alpha(g, \psi) = M\alpha = (E - \psi G)^{-1}\alpha. \tag{6}$$

In Katz–Bonacich network centrality, matrix $M$ is expressed as

$$M = (E - \psi G)^{-1} = E + \psi G + \psi^2 G^2 + \ldots \tag{7}$$

$M$ can be regarded as a social multiplier, which is the key mechanism to generate network effects. It can reflect the cascade characteristics and attenuation characteristics of different individuals in the network. In social networks, the behavior choices of individuals are influenced by neighbors, enabling their determination. Therefore, this mutual influence mechanism is persistent in the network until it reaches convergence under certain conditions.

In the game, local governments simultaneously choose their own agricultural carbon peak level to maximize their respective income functions. The optimal response function can be obtained from the first-order optimization conditions.

$$c_i^*(\boldsymbol{Env}, \boldsymbol{g}) = \sigma \alpha_i + \zeta_i(\boldsymbol{Env}) + \psi \sum_{j=1}^{n} g_{ij} c_j \tag{8}$$

Substituting Equation (2) into Equation (8) yields

$$c_i^*(\boldsymbol{Env}, \boldsymbol{g}) = \sigma \alpha_i + \psi \sum_{j=1}^{n} g_{ij} c_j + \beta_0 Env_i + \frac{1}{d_i(\boldsymbol{g})} \gamma_0 \sum_{j}^{n} g_{ij} Env_j, \tag{9}$$

which can be written in matrix form as $\boldsymbol{c}^* = [\sigma \alpha + \boldsymbol{\zeta}] + \psi \boldsymbol{G} \boldsymbol{y}$. The carbon peak level of the game equilibrium can be obtained by solving the following equation:

$$\boldsymbol{c}^* = [E - \psi \boldsymbol{G}]^{-1} [\sigma \alpha + \boldsymbol{\zeta}] = M[\sigma \alpha + \boldsymbol{\zeta}], \tag{10}$$

where $\omega(\boldsymbol{g})$ is the maximum eigenvalue of the network adjacency matrix $\boldsymbol{G} = [g_{ij}]$. Then, the following proposition can be obtained: if $\psi \omega(\boldsymbol{g}) < 1$, there is a unique Nash equilibrium, and the equilibrium result is equal to the centrality of the corresponding weighted Katz–Bonacich network, i.e.,

$$c_i^*(\boldsymbol{Env}, \boldsymbol{g}) = b_{(\sigma \alpha + \zeta)i}(\boldsymbol{g}, \psi). \tag{11}$$

For the proof of this proposition, please refer to the appendix at the end of Helsley et al. [45]. The expression of the equilibrium agricultural carbon peak level of local government $i$ can be specifically expanded as follows:

$$c_i^*(\boldsymbol{Env}, \boldsymbol{g}) = \sum_{j=1}^{n} \sum_{k=0}^{+\infty} \psi^k g_{ij}^{[k]} [\sigma \alpha_j + \zeta_j(\boldsymbol{Env})]. \tag{12}$$

It can be seen that the peak level of agricultural carbon in the Nash equilibrium depends on the network adjacency matrix and regional heterogeneity factors. The above formula also shows that, if a local government occupies a more dominant position in the network, i.e., if it has a greater Katz–Bonacich network centrality, it will also tend to choose a higher agricultural carbon peak level. In addition, when the impact of regional heterogeneity factors on environmental governance is positive, the peak level of balanced agricultural carbon will also increase. Obviously, $\psi$ reflects the degree of strategic interaction between local governments, in addition to depicting the degree of strategic complementarity and competitive interaction in the network. An increase in $\psi$ will significantly increase the agricultural carbon peak level of all local governments.

According to the theoretical model, $\psi$ is internally consistent with the spatial dependence of local government competition, as discussed later in this paper. The spatial econometric model incorporates the action mechanism of the spatial autoregressive coefficient or spatial dependence parameter and the network game $\psi$. The spatial weight matrix is equivalent to the adjacency matrix in the network game model, which provides an important basis for this paper to organically combine the network game model of local



government competition with the spatial measurement model. In the demonstration of the spatial econometric model, this paper builds the corresponding estimation equation as a function of the expression of the optimal response function of the local government's agricultural carbon peak goal before conducting the corresponding empirical test.

## 3. Spatial Measurement Model Settings, Data Source, and Variable Description

### 3.1. Model Settings

On the basis of the theoretical analysis results of the network game model, this paper uses the spatial econometric model to test the impact mechanism of environmental governance and local government competition on the peak level of agricultural carbon. The network relationship weight $G$ in the network game model has a certain commonness with the $W$ in the spatial econometric model, and the social multiplier and the spatial multiplier have high similarity and internal correlation. According to the optimal response function of the local government, the spatial panel model set in this paper is

$$carbon_{it} = \rho \sum_{j=1}^{N} w_{ij} carbon_{jt} + \beta_0 Env_{it} + \gamma_0 \sum_{j=1}^{N} w_{ij} Env_{jt} + \sum_{h=1}^{H} \beta_h x_{it}^h + \sum_{h=1}^{H'} \sum_{j=1}^{N} \gamma_h w_{ij} x_{jt}^h + \mu_i + \varepsilon_{it}, \quad (13)$$

where $i$ and $j$ represent different provinces, $t$ represents each year, $carbon_{it}$ represents the peak level of agricultural carbon in period $t$ of the $i$th province, $carbon_{jt}$ represents the peak level of agricultural carbon corresponding to the competing provinces, $x_{it}$ is the control variable that affects the peak level of agricultural carbon, $Env_{it}$ represents the environmental governance of the $i$th province in period $t$, $Env_{jt}$ represents the environmental governance corresponding to the competitive provinces, $\mu_i$ is the interprovincial individual effect, $\varepsilon_{it}$ is a random disturbance term, and $w_{ij}$ is the key spatial weight in the model. In the subsequent analysis, the spatial adjacency matrix is used as the basic regression, and the distance weight matrix, the distance square weight matrix, and the economic distance weight matrix are used for robustness analysis. As expressed in Equation (13), spatial autoregressive coefficients $\rho$, $\beta_0$, and $\gamma_0$ are the core parameters of this study. If $\rho$ is significantly greater than 0, it indicates that local government competition has strategic complementary behavior; if $\rho$ is significantly less than 0, it indicates that there is strategic substitution behavior in local government competition.

### 3.2. Data Source and Variable Description

Considering the availability and accuracy of the research content and data, this paper selects the data of various provinces and regions in China from 2005 to 2020. Due to the lack of data in Tibet, it was excluded, and a total of 480 research samples from 30 provincial-level regions were obtained. Relevant research data were obtained from the China fiscal Yearbook [46], EPS data platform [47], China Energy Statistical Yearbook [48], China Rural Statistical Yearbook [49], and China fiscal Yearbook [50].

#### 3.2.1. Description of the Dependent Variable

The explained variable in the measurement model was the agricultural carbon peak level, which is expressed by subtracting the ratio of the agricultural carbon emission intensity value of each province and the agricultural carbon emission intensity value under the carbon peak state from 1. When the carbon reaches the peak, it means that at a certain time point, carbon dioxide emissions will no longer increase to the peak, and then gradually fall back. According to this background, this paper uses 35% of the national agricultural carbon emission intensity value in 2005 as the agricultural carbon emission intensity value in the carbon peak state. The agricultural carbon emission intensity used in this paper is the ratio of agricultural carbon emissions to the added value of the primary industry. The added value of the primary industry is adjusted by using the added value index of the primary industry in the base period of 2005. See the research of Tian Y. et al. [51] for the specific measurement method of agricultural carbon emissions. According to the definition, a greater peak level of agricultural carbon indicates greater achievements in agricultural

carbon emission reduction. When the peak level of agricultural carbon is less than zero, the intensity of agricultural carbon emission in this region has not reached the peak state; when the peak level of agricultural carbon is equal to zero, the intensity of agricultural carbon emission in the region has reached the peak state of carbon; when the peak level of agricultural carbon is greater than zero, the agricultural carbon emission intensity in this region has not only reached the expected peak state of carbon but is also further striving to achieve the goal of carbon neutralization.

3.2.2. Description of Independent Variables

The core explanatory variable used in the model was environmental governance (envpro). The calculation of environmental governance variables referred to the research of Chen S. et al. [52], whereby the government work reports of 30 provincial-level areas in the Chinese Mainland were manually collected from 2005 to 2020, before conducting word segmentation processing and statistical analysis. The frequency of words related to the environment in the provincial-level government work reports accounted for the total number of words in the full text of the government work reports to represent the strength of the government's environmental governance.

Other control variables were the agricultural industrial structure (ainstru), agricultural public investment (pubinvestments), fiscal decentralization (fisexp), regional industrial structure (primarypro), and rural population proportion (rupoppro). The agricultural industrial structure was expressed by the proportion of the total output value of planting and animal husbandry in the total output value of agriculture, forestry, animal husbandry, and fishery, mainly considering that the carbon emissions generated by planting and animal husbandry in production activities are greater than those of other agricultural industrial sectors [53]. The public investment in agriculture was expressed by the investment amount of fixed assets in agriculture, forestry, animal husbandry, and fishery. Previous study has shown that an increase in fixed assets investment in agriculture, forestry, animal husbandry, and fishery plays a certain role in inhibiting agricultural carbon emissions [54]. Fiscal decentralization used a decentralized structure at the level of fiscal expenditure, expressed by the ratio of per capita provincial fiscal expenditure to the sum of per capita provincial fiscal expenditure and per capita central fiscal expenditure. The regional industrial structure was expressed by the proportion of the added value of the primary industry in the regional GDP. The proportion of the rural population was expressed by the proportion of the total rural population of the region to the total population of the region. The descriptive statistical results of each variable are shown in Table 1.

**Table 1.** Variable description and descriptive statistics.

| Variable Name | Obs. | Measure | Mean | SD [a] | MIN | MAX |
|---|---|---|---|---|---|---|
| carbon | 480 | — | −1.462 | 1.453 | −10.643 | 0.318 |
| envpro | 480 | % | 0.600 | 0.253 | 0.078 | 1.529 |
| ainstru | 480 | % | 0.825 | 0.104 | 0.540 | 0.960 |
| fisexp | 480 | — | 0.510 | 0.122 | 0.202 | 0.937 |
| pubinves | 480 | 100 million CNY | 467.800 | 574.597 | 1.100 | 3814.470 |
| primarypro | 480 | % | 10.714 | 5.740 | 0.300 | 33.700 |
| rupoppro | 480 | % | 44.835 | 14.008 | 10.417 | 73.137 |

[a] SD = standard deviation.

## 4. Analysis of Empirical Results

### 4.1. Basic Regression Results

Table 2 shows the basic regression results of estimation based on spatial adjacency weight. Model 1 shows the estimation results of the spatial panel SAR model while model 2 shows the estimation results of the SDM model. In this paper, the likelihood ratio test (LR test) was used to compare and select the SAR model and the SDM model. The test results show that the likelihood ratio statistic was 36.320, and the corresponding $p$-value was 0.000.

Therefore, SAR could be rejected as a nested model of SDM, and the SDM model should be selected. In addition, this paper also used the Hausman test to verify the random and fixed effects of the spatial panel model. It can be seen from Table 2 that, in the SDM model, the *p*-value of the Hausman test was 0.000, rejecting the original hypothesis and choosing the fixed effect. The subsequent analysis of this paper is based on the fixed-effect panel SDM model. From the estimation results of model 2, it can be seen that the spatial autoregressive coefficient reflecting the strategic interaction of local governments was 0.131, and it was positive at the significance level of 5%, which indicates that local governments have positive competition with respect to the agricultural carbon peak, and they adopt a complementary carbon peak competition strategy. That is, when the carbon peak level of the surrounding provinces increases, the carbon peak level of the local region will also increase. There is a phenomenon of mutual imitation and convergence between the neighboring provinces. The root of this kind of competition strategy behavior lies in the driving force of local government competition incentives under the decentralization system, and this kind of competition mainly stems from political incentives. Local governments compete with each other strategically in order to gain advantages in performance assessment. This internal motivation drives them to attach importance to the agricultural carbon emission intensity index, as well as strive to improve the local carbon peak level. Therefore, under the decentralized system, the competition between local governments strengthens the local government's carbon reduction and emission reduction behavior.

**Table 2.** Strategic interaction estimation of local government's carbon peak goal.

| Variable Name | Model 1 SAR | | | Model 2 SDM | | |
|---|---|---|---|---|---|---|
| | Coefficient | SE [a] | Z-Statistic | Coefficient | SE [a] | Z-Statistic |
| ρ | 0.163 *** | 0.049 | 3.320 | 0.131 ** | 0.059 | 2.200 |
| envpro | 0.451 *** | 0.076 | 5.910 | 0.467 *** | 0.077 | 6.040 |
| ainstru | −1.212 ** | 0.475 | −2.550 | −1.035 * | 0.537 | −1.930 |
| fisexp | 0.188 | 0.446 | 0.420 | 1.256 ** | 0.521 | 2.410 |
| pubinves | −0.000 *** | 0.000 | −5.590 | −0.000 *** | 0.000 | −5.680 |
| primarypro | 0.022 *** | 0.008 | 2.630 | 0.014 * | 0.008 | 1.700 |
| rupoppro | −0.067 *** | 0.005 | −13.370 | −0.050 *** | 0.008 | −6.360 |
| W × envpro | | | | −0.021 | 0.128 | −0.170 |
| W × ainstru | | | | 3.515 *** | 0.838 | 4.200 |
| W × fisexp | | | | 2.234 * | 1.262 | 1.770 |
| W × pubinves | | | | 0.000 *** | 0.000 | 2.900 |
| W × primarypro | | | | −0.036 ** | 0.016 | −2.300 |
| W × rupoppro | | | | −0.004 | 0.010 | −0.390 |
| Obs. | 480 | | | 480 | | |
| Hausman test | 10.530 | | | 38.070 | | |
| Hausman *p*-value | 0.160 | | | 0.000 | | |
| LR Test χ² | | | 36.320 | | | |
| *p* | | | 0.000 | | | |

Note: ***, **, and * denote statistical significance at the 1%, 5%, and 10% levels, respectively. [a] SE = standard error.

In Table 2, each independent variable coefficient of model 2 indicates the influence of each independent variable on the local agricultural carbon peak level. For the interpretation of the estimated coefficient of each independent variable, the conventional estimation coefficient interpretation method cannot be directly applied. It is necessary to further calculate the direct effect, indirect effect, and total effect of the relevant independent variables. The results are shown in Table 3. The direct effect in Table 3 is the sum of the spatial Dubin model coefficient and the feedback effect. The feedback effect indicates that the independent variable of a certain region will have an impact on the agricultural carbon peak level of its surrounding provinces, which, in turn, will affect the agricultural carbon

peak level of the region, which is also called the "regional spillover effect". The indirect effect is also called the "spatial spillover effect", indicating the impact of an independent variable of the surrounding provinces on the peak level of agricultural carbon in this region. The total effect is the sum of the direct effect and indirect effect, indicating the average impact of the change in an independent variable in a certain region on the peak level of agricultural carbon in all regions. By combining the results in Tables 2 and 3, the results of the respective variables can be explained in detail.

**Table 3.** Calculation results of direct and indirect effects.

| Variable Name | LR_Direct | | LR_Indirect | | LR_Total | |
|---|---|---|---|---|---|---|
| | Coefficient | Z-Statistic | Coefficient | Z-Statistic | Coefficient | Z-Statistic |
| envpro | 0.468 *** | 6.390 | 0.072 ** | 2.530 | 0.540 *** | 6.410 |
| ainstru | −0.957 * | −1.800 | 3.652 *** | 3.940 | 2.695 *** | 2.620 |
| fisexp | 1.291 ** | 2.510 | 2.665 ** | 2.040 | 3.956 ** | 2.430 |
| pubinves | −0.000 *** | −5.700 | 0.000 *** | 3.000 | −0.000 | −0.170 |
| primarypro | 0.013 | 1.640 | −0.037 ** | −2.280 | −0.024 | −1.360 |
| rupoppro | −0.051 *** | −8.210 | −0.008 *** | −2.800 | −0.059 *** | −9.160 |

Note: ***, **, and * denote statistical significance at the 1%, 5%, and 10% levels, respectively.

The spatial Dubin regression coefficient of environmental governance (envpro) in Table 2 was 0.467, showing a significant positive relationship with the local agricultural carbon peak level, indicating that a stronger environmental governance capacity is more conducive to the improvement of the local agricultural carbon peak level. In Table 3, the direct effect coefficient value of environmental governance (envpro) was 0.468, with a feedback effect of 0.002, indicating that a stronger local environmental governance ability is conducive to the improvement of the agricultural carbon peak level of neighboring provinces, and this impact will, in turn, promote the improvement of the local agricultural carbon peak level. The indirect effect coefficient of environmental governance (envpro) was 0.072, indicating that the environmental governance of neighboring provinces has a significant positive relationship with the local agricultural carbon peak level, along with a significant spatial spillover effect, whereby increasing environmental governance in neighboring regions is conducive to the improvement of the local agricultural carbon peak level. The direct effect coefficient of environmental governance (envpro) was significantly greater than the indirect effect coefficient, indicating that the impact of local environmental governance on the peak level of agricultural carbon is significantly greater than the impact of environmental governance of neighboring provinces on the peak level of local agricultural carbon. According to the coefficient of total effect, environmental governance (envpro) has a significant positive average impact on the peak level of agricultural carbon in all regions, whereby increasing environmental governance is conducive to the improvement of the peak level of agricultural carbon in all regions.

The spatial Dubin regression coefficient of the agricultural industrial structure (ainstru) was −1.085, which was significantly negatively related to the local agricultural carbon peak level. That is, a greater proportion of the total output value of local animal husbandry and planting industry in the output value of agriculture, forestry, animal husbandry, and fishery leads to greater agricultural carbon emissions and a more unfavorable promotion of the local agricultural carbon peak level. The indirect effect coefficient of the agricultural industrial structure (ainstru) was 3.652, passing the significance test at the level of 1%, indicating that the agricultural industrial structure has a significant spatial spillover effect on the peak level of agricultural carbon; i.e., there is a significant positive relationship between the agricultural industrial structure of adjacent areas and the local agricultural carbon peak level. According to the coefficient and direction of the total effect, the agricultural industrial structure (ainstru) has a significant positive average impact on the peak level of agricultural carbon in all regions.

The spatial Dubin regression coefficient of fiscal decentralization (fisexp) was 1.230, showing a significant positive relationship with the local agricultural carbon peak level, indicating that a higher local fiscal expenditure decentralization is more conducive to the improvement of the local agricultural carbon peak level. The indirect effect coefficient of fiscal expenditure decentralization (fisexp) was 2.665, indicating that the fiscal decentralization level of the neighboring provinces has a significant positive relationship with the local agricultural carbon peak level, along with a significant spatial spillover effect; i.e., a higher fiscal decentralization level in neighboring regions results in a better local agricultural carbon peak level. According to the coefficient and direction of the total effect, fiscal decentralization (fisexp) has a significant positive average impact on the peak level of agricultural carbon in all regions, whereby a higher degree of fiscal decentralization is more conducive to the improvement of the peak level of agricultural carbon in all regions.

The spatial Dubin regression coefficient of agricultural public investment (pubinves) was −0.000, which had a significant negative relationship with the local agricultural carbon peak level. That is, a greater local investment in fixed assets of agriculture, forestry, animal husbandry, and fishery is more unfavorable to the improvement of the local agricultural carbon peak level. A possible explanation is that fixed asset investment activities will cause additional carbon emissions, and, because the return period is long, this has little effect on the increase in agricultural output value in the short term. The direct effect coefficient of agricultural public investment (pubinves) indicates that the local agricultural public investment will further affect the realization of the local agricultural carbon peak target under the influence of a feedback effect, while the indirect effect coefficient indicates that the agricultural public investment of neighboring provinces has a positive impact on the agricultural carbon peak level of the region. Under the offset of direct and indirect effects, the negative average impact of agricultural public investment (pubinves) on the peak level of agricultural carbon in all regions was not significant.

The spatial Dubin regression coefficient of the regional industrial structure (primarypro) was 0.014, which had a positive relationship with the local agricultural carbon peak level, passing the significance test at the level of 10%. The direct effect coefficient of the regional industrial structure (primarypro) shows that the positive relationship between the proportion of the primary industry and the peak level of agricultural carbon was not significant. The indirect effect coefficient of the regional industrial structure (primarypro) was −0.037, indicating that the regional industrial structure of the neighboring provinces has a significantly negative relationship with the local agricultural carbon peak level, along with a significant spatial spillover effect, whereby the reduction in the proportion of the primary industry in the neighboring regions is conducive to the improvement of the local agricultural carbon peak level. Agriculture itself has two attributes with respect to carbon sinks and carbon emissions. Under the offset of direct and indirect effects, the negative average impact of the regional industrial structure (primarypro) on the peak level of agricultural carbon in all regions was not significant.

The spatial Dubin regression coefficient of the proportion of rural population (rupoppro) was −0.051, showing a significant negative relationship with the local agricultural carbon peak level; i.e., a smaller proportion of the local rural population is more conducive to the improvement of the local agricultural carbon peak level. The indirect effect coefficient indicates that the proportion of rural population (rupoppro) in the neighboring provinces has a significant spatial spillover effect on the peak level of agricultural carbon in the region, whereby a reduction in the proportion of the rural population in the neighboring provinces brings about an increase in the peak level of local agricultural carbon. According to the coefficient and direction of the total effect, the proportion of the rural population(rupoppro) has a significant negative average impact on the peak level of agricultural carbon in all regions, whereby a reduction in the proportion of the rural population is conducive to the improvement of the peak level of agricultural carbon in all regions. Generally speaking, the provinces with a high proportion of rural population are mostly large agricultural

provinces, and the agricultural carbon emissions are relatively high; hence, the peak level of agricultural carbon is lower than that of other provinces.

*4.2. Robustness Analysis of Different Spatial Weights*

In order to verify the robustness and reliability of the model estimation results, this paper used the spatial distance weight matrix, the distance square weight matrix, and the economic distance weight matrix to carry out regression analysis on the model. The setting method of each weight matrix form was previously described by Peng Xuhui et al. [44]. The estimation results are shown in Table 4. The Hausman test results show that the panel SDM model using fixed effects was supported under the three spatial weight matrices. Regardless of the spatial weight matrix used, the spatial autoregressive coefficient of agricultural carbon peak level was positive at the significance level of 1%. Local governments have obvious strategic complementary behaviors in competition with respect to the agricultural carbon peak level, and the interaction effect of this competition is very stable. According to the regression coefficient value, the spatial autoregressive coefficient based on the distance space weight matrix and the distance square space weight matrix is relatively large. This indicates that geographical factors are still the main factors to be considered in the local government's agricultural carbon peak strategy. When the local government interacts with the agricultural carbon peak strategy, it is still more inclined to take geographically adjacent regions as the yardstick. The spatial Dubin regression coefficient of environmental governance (envpro) changed little, and the results were also very stable.

**Table 4.** Strategic interaction estimation of local governments' carbon peak goals under different spatial weight matrices.

| Variable Name | Weight of Distance | | Weight of Distance Square | | Weight of Economic Distance | |
|---|---|---|---|---|---|---|
| | Coefficient | Z-Statistic | Coefficient | Z-Statistic | Coefficient | Z-Statistic |
| ρ | 0.395 *** | 5.600 | 0.277 *** | 3.720 | 0.136 *** | 2.590 |
| envpro | 0.407 *** | 5.450 | 0.399 *** | 5.350 | 0.443 *** | 6.040 |
| ainstru | −1.676 *** | −3.440 | −1.551 *** | −3.180 | −2.494 *** | −5.070 |
| fisexp | 0.202 | 0.460 | 0.553 | 1.190 | −0.254 | −0.590 |
| pubinves | −0.000 *** | −5.110 | −0.000 *** | −5.350 | −0.000 *** | −6.050 |
| primarypro | 0.028 *** | 3.410 | 0.029 *** | 3.550 | 0.033 *** | 4.100 |
| rupoppro | −0.065 *** | −10.910 | −0.053 *** | −7.380 | −0.060 *** | −10.870 |
| Hausman test | 17.960 | | 16.880 | | 21.960 | |
| Hausman *p*-value | 0.022 | | 0.051 | | 0.015 | |
| Obs. | 480 | | 480 | | 480 | |

Note: *** denotes statistical significance at the 1% level.

This paper also calculated the direct effect, indirect effect, and total effect of environmental governance and other control variables under different spatial weight matrices. The results are shown in Table 5. Under the three weight matrices, the direct effect, indirect effect, and total effect coefficient of environmental governance (envpro) were significantly positive. In addition, agricultural industrial structure (ainstru) and fiscal decentralization (fisexp) had a significant positive average impact on the peak level of agricultural carbon, and agricultural public investment (pubinves) and rural population proportion (rupoppro) had a significant negative average impact on the peak level of agricultural carbon. These results are in good agreement with the calculation results based on spatial adjacency weight, further indicating that the research conclusions of this paper are robust and that changes in the spatial weight matrix would not affect the main research conclusions of this paper.

**Table 5.** Calculation results of direct and indirect effects under different spatial weight matrices.

| Variable Name | | Envpro | Ainstru | Fisexp | Pubinves | Primarypro | Rupoppro |
|---|---|---|---|---|---|---|---|
| Weight of distance | LR_Direct | 0.413 *** | −1.505 *** | 0.189 | −0.000 *** | 0.028 *** | −0.065 *** |
| | LR_Indirect | 0.271 *** | 8.363 *** | 0.142 | −0.000 *** | 0.019 ** | −0.043 *** |
| | LR_Total | 0.683 *** | 6.858 *** | 0.331 | −0.000 *** | 0.047 *** | −0.108 *** |
| Weight of distance square | LR_Direct | 0.404 *** | −1.401 *** | 0.544 | −0.000 *** | 0.029 *** | −0.054 *** |
| | LR_Indirect | 0.152 ** | 4.018 *** | 0.204 | −0.000 ** | 0.011 ** | −0.044 *** |
| | LR_Total | 0.556 *** | 2.617 ** | 0.747 | −0.000 *** | 0.040 *** | −0.098 *** |
| Weight of economic distance | LR_Direct | 0.445 *** | −2.339 *** | −0.200 | −0.000 *** | 0.032 *** | −0.060 *** |
| | LR_Indirect | 0.072 ** | 5.327 *** | 2.682 ** | −0.000 ** | −0.071 *** | −0.010 ** |
| | LR_Total | 0.518 *** | 2.988 *** | 2.482 * | −0.000 *** | −0.040 ** | −0.070 *** |

Note: ***, **, and * denote statistical significance at the 1%, 5%, and 10% levels, respectively.

## 5. Discussion

This paper discussed the issue of agricultural carbon emissions in China under the background of introducing the goal of carbon peaking, which used the network game model to analyze the impact of environmental governance on the agricultural carbon peaking level from the theoretical level and the strategic interaction between local governments on the goal of agricultural carbon peaking, and we used the spatial econometric model to empirically test the conclusions drawn from the theoretical analysis. The theoretical and empirical analysis of this paper shows that environmental governance and local government competition play important roles in achieving the goal of agricultural carbon peak [11,15,17]. Previous studies have shown that there is a significant spatial correlation between carbon emissions [36,38,51,55–57], which indicates that carbon reduction and emission reduction cannot rely on the unilateral actions of various regions [43,58]. Local governments have positive competition in the competition for agricultural carbon peak, and there is mutual imitation and convergence between neighboring provinces [59,60]. Under the recognition of the common goal of reaching the carbon peak, all regions should strengthen environmental governance and attach importance to carbon emission reduction cooperation to promote carbon emission reduction at a lower cost [53,61–64].

Compared with the existing studies, the main contributions of this paper are reflected in two aspects. Firstly, most of the existing studies on agricultural carbon emissions are biased toward the construction of indicator systems and quantitative measurement analysis, with less focus on the government behavior driving factors behind agricultural carbon emissions. In this paper, on the basis of existing research, the driving mechanism and influencing factors of the agricultural carbon peak were systematically analyzed. Secondly, this paper creatively introduces the network game theory to analyze the impact of environmental governance on the agricultural carbon peak level and the effect of strategic interaction behavior among local governments on the agricultural carbon peak target. On the basis of theoretical analysis, these impacts are verified through the spatial econometric model, which represents an innovative approach in the literature.

It is undeniable that there are still some deficiencies in the theoretical analysis and empirical analysis of this work. In fact, straw burning is also one of the sources of agricultural carbon emissions [65]. However, it is difficult to estimate the quantity of straw burning accurately. Due to data limitations, this paper does not consider the carbon emissions caused by the open burning of crop residues when calculating the agricultural carbon emissions of various provinces in China. However, the agricultural carbon emission measurement system used in this paper fully considers the carbon emissions caused by animal breeding, rice planting, and energy input in agricultural production, which is scientific and reasonable. In the future, it is planned to further improve the measurement system of agricultural carbon emissions to reduce the error of research results.

## 6. Conclusions and Recommendations

### 6.1. Main Conclusions

This paper described the internal relationship among environmental governance, local government competition, and the peak level of agricultural carbon by introducing the network game model, as well as empirically analyzing the strategic interaction between local governments using the data of 30 provincial-level regions in the Chinese Mainland from 2005 to 2020 combined with the spatial measurement panel model, thus realizing the organic integration of theoretical analysis and empirical testing. The empirical analysis based on the spatial Dubin panel model found that there is a positive competition among local governments in the competition with respect to agricultural carbon peaks, and they adopt a complementary carbon peak competition strategy. That is, when the carbon peak level of the surrounding provinces increases, the carbon peak level of the local region will also increase. There is a phenomenon of mutual imitation and convergence between the neighboring provinces. Under different spatial weight matrix settings, the competitive interaction effects of local governments in agricultural carbon peak are stable and reliable, and the local governments are more inclined to take geographically adjacent regions as the yardstick in the competition with respect to agricultural carbon peak strategy. Secondly, environmental governance has a significant positive direct effect and a positive spatial spillover effect. Increasing environmental governance is conducive to the improvement of the local agricultural carbon peak level. Increasing environmental governance in neighboring provinces can also promote the improvement of the local agricultural carbon peak level. From the perspective of the coefficients, its direct effect is significantly greater than the spatial spillover effect. Therefore, more attention should be paid to local environmental governance to promote the improvement of the agricultural carbon peak level. Thirdly, agricultural industrial structures, fiscal decentralization, agricultural public investment, environmental governance, regional industrial structure, and the proportion of the rural population have significant spatial spillover effects. Agricultural industrial structure and fiscal decentralization are significantly positively correlated with the peak level of agricultural carbon, while the proportion of the rural population is significantly negatively correlated with the peak level of agricultural carbon.

### 6.2. Policy Implications

The theoretical and empirical analysis of this paper showed that environmental governance and local government competition play important roles in achieving the goal of agricultural carbon peak. In order to better achieve the goal of agricultural carbon peak and promote the high-quality development of agriculture, on the basis of the above research conclusions, this paper puts forward some policy recommendations.

Firstly, the cooperation and exchange of local governments in agricultural carbon emission reduction and the collaborative governance capacity of regional agricultural carbon emissions should be strengthened. An information transmission platform should be built for regional agricultural carbon emission control, and cooperation and exchanges among various regions should be strengthened, especially in neighboring regions. On the basis of considering the differences in the total amount and sources of agricultural carbon emissions in various regions, the emission reduction advantages of various regions can be considered. In the deployment of agricultural carbon emission policies, the interactive factors of spatial strategies, the demonstration role of typical regions, and the imitation of surrounding regions should be fully considered, while emphasizing regional linkage to improve the collaborative governance ability to reduce regional agricultural carbon emissions.

Secondly, efforts to improve agricultural environmental protection should be intensified. The mode of agricultural development should be changed while implementing the action of agricultural green development. The prevention and control mechanism of agricultural non-point source pollution should be improved while increasing investment in the treatment and restoration technology of polluted farmland, as well as improving the

utilization rate of chemical fertilizers and pesticides for crops. Collection points should be set up for pesticide packaging wastes while exploring multiple ways to recover them. Farmers should be guided to discard agricultural film and other production wastes to avoid "white pollution". The resource utilization of livestock and poultry breeding wastes should be accelerated while improving the supervision of livestock and poultry breeding pollution, as well as minimizing the pollution impact caused by livestock and poultry breeding wastes.

Thirdly, the agricultural industrial structure should be adjusted and optimized to a green and low-carbon transformation. On the basis of adhering to the bottom line of food security, the agricultural production structure should be adjusted and optimized while improving the level of agricultural industrialization, specialization, and agglomeration, as well as the agricultural production efficiency. The transformation of the agricultural production mode should be actively ushered from the traditional production mode of "high energy consumption, high emissions, high pollution, and low carbon sink" to the modern, low-carbon agricultural production mode of "low energy consumption, low emissions, low pollution, and high carbon sink". Investment in green and ecological agriculture should be increased while improving agricultural production infrastructure. The "three products and one standard" certification and brand building of agricultural products should be accelerated while improving the quality and popularity of local agricultural products. New modes and new paths of green agriculture whole-chain operation and management should be explored while extending the industrial chain, as well as improving the driving ability of the industrialized operation mode.

Fourthly, the role of fiscal policy should be fully considered in supporting and guiding the development of low-carbon agriculture. Financial input should be increased while giving appropriate policy preference to the development of green and low-carbon agriculture. Furthermore, the structure of financial subsidies for agriculture should be adjusted while guiding the vast number of agricultural practitioners to adopt low-carbon production methods through financial means, allowing them to effectively participate in the protection of arable land resources and ecological environment while constantly cultivating their habits of low-carbon production and low-carbon consumption.

Lastly, the efficiency of agricultural public investment should be improved. The structure of public investment in agriculture should be continuously optimized while increasing investment in agricultural infrastructure. The construction of agricultural projects such as high-standard farmland, the storage and preservation of agricultural products, and cold-chain logistics should be accelerated while constantly improving agricultural production conditions. The environment for public investment in agriculture should be improved, and the ability of regions to absorb public investment in agriculture should be enhanced. The supervision and regulation of agricultural public investment funds should be strengthened while constantly improving the efficiency of agricultural public investment.

**Author Contributions:** Conceptualization, Y.Y., X.P., and Y.T.; methodology, Y.Y. and X.P.; software, Y.Y., H.Y., and X.P.; formal analysis, Y.Y. and X.P.; resources, W.W.; data curation, Y.Y., H.Y., and M.Y.; writing—original draft preparation, Y.Y.; writing—review and editing, Y.T., X.P., and M.Y.; visualization, Y.Y.; supervision, Y.Y.; funding acquisition, Y.Y. All authors have read and agreed to the published version of the manuscript.

**Funding:** This research was funded by the key scientific research projects of colleges and universities in Henan Province (no. 23A790025), National natural science foundation of China (no. 71903197; no. 71804059), Anyang science and technology plan soft science research project (no. 2022C02ZF013), and the Doctoral research startup fund project of Anyang Institute of Technology (no. BSJ2020001).

**Institutional Review Board Statement:** By convention, the Institutional Review Board (IRB) statement is exempted because (1) one of the authors has the job responsibility of collecting and handling the data; (2) the analysis was conducted on data anonymized by that author and thus imposed no risk; (3) our institutions have no IRB.

**Informed Consent Statement:** Not applicable.

**Data Availability Statement:** The data that support the findings of this study are available from the authors upon reasonable request.

**Conflicts of Interest:** The authors declare no conflict of interest.

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
