# Peer review of "Research on Environmental Governance, Local Government Competition, and Agricultural Carbon Emissions under the Goal of Carbon Peak"

_agriculture, doi:10.3390/agriculture12101703_

Round 1

Reviewer 1 Report

The proposed research « Research on Environmental Governance, Local Government Competition, and Agricultural Carbon Emissions under the Goal of Carbon Peak» falls within the scope of Agriculture. According to the reviewer’s opinion, minor revisions are required in order to accept this research study for publication Agriculture. Please, comply with the following suggestions and comments:

Comment 1: The paper is in general well accompanied of clear explanations. I think that some additional figures would help to the better analysis of the subject.

Comment 2: More recent papers in the field should be integrated in the literature review.

Comment 3: Finally, when you submit the corrected version, please do check thoroughly, in order to avoid grammar, syntax or structure/presentation flaws - please seek for professional English proofreading services or ask a native English-speaking colleague of yours in order to refine and improve the English in your paper.

Author Response

To reviewer 1:

Dear reviewer and editor,

Thank you very much for your valuable advice. Your opinions are very important. We have carefully revised the following according to the review opinions of the paper.

Comment 1: The paper is in general well accompanied of clear explanations. I think that some additional figures would help to the better analysis of the subject.

Thank you for your valuable advice. According to your suggestion, we have added explanations to some contents.

Comment 2: More recent papers in the field should be integrated in the literature review. 

Thank you for your valuable advice. According to your suggestion, we have added some recent papers in this field.

Comment 3: Finally, when you submit the corrected version, please do check thoroughly, in order to avoid grammar, syntax or structure/presentation flaws - please seek for professional English proofreading services or ask a native English-speaking colleague of yours in order to refine and improve the English in your paper.

Thank you for your valuable advice. According to your suggestion, we have carefully checked and checked the grammar.

Reviewer 2 Report

In general paper is well written and structured. However, there are some points, which must be adressed.

1) Main conclusions as well as policy implications must be properly discussed and justified based on existing research. Authors must properly place their results in existig scientific field.

2) maybe it is better to add proper xonxlusion, which will briefly state main results, limitations and further directions of the research. 

Author Response

To reviewer 2:

Dear reviewer and editor,

Thank you very much for your valuable advice. Your opinions are very important. We have carefully revised the following according to the review opinions of the paper. We have revised the normative of references and references.

Point 1: Main conclusions as well as policy implications must be properly discussed and justified based on existing research. Authors must properly place their results in existig scientific field.

Thank you for your valuable advice. According to your suggestion,We have discussed the theoretical contribution and practical application of the conclusions in detail.

Point 2: maybe it is better to add proper xonxlusion, which will briefly state main results, limitations and further directions of the research. 

Thank you for your valuable advice. According to your suggestion , we have added the discussion part in the paper to supplement the main results, limitations and further directions of the paper.

Reviewer 3 Report

While the article proposal is interesting I have some concerns regarding the assumptions:

1. My major concern with the article is the assumption that all provinces have the same priorities in terms of optimizing the agriculture sector (primary industry). It is clear that some provinces put more emphasis in the secondary industry (manufacturing) or tertiary industry.

2. In the case of the agriculture sector in China, the study by Liang et al (2021) found that most of the GHG emissions from crop sector come from crop residue open burning, rice cultivation and cropland emissions account for most of it. The study should also consider this fact

https://www.nature.com/articles/s41597-021-00960-5#Sec10 

Author Response

To reviewer 3:

Dear reviewer and editor,

Thank you very much for your valuable advice. Your opinions are very important. We have carefully revised the following according to the review opinions of the paper.

Point 1: My major concern with the article is the assumption that all provinces have the same priorities in terms of optimizing the agriculture sector (primary industry). It is clear that some provinces put more emphasis in the secondary industry (manufacturing) or tertiary industry.

Thank you for your valuable advice. Your opinion is very pertinent and we have benefited a lot. Different regions have different agricultural carbon emissions, different emphasis on agricultural carbon emissions, and different deployment plans in agricultural carbon emission reduction. We believe that on the premise of realizing the common goal of carbon peak, all regions should strengthen environmental governance, attach importance to carbon emission reduction cooperation, and promote carbon emission reduction at a lower cost.

Point 2: In the case of the agriculture sector in China, the study by Liang et al (2021) found that most of the GHG emissions from crop sector come from crop residue open burning, rice cultivation and cropland emissions account for most of it. The study should also consider this fact.

Thank you for your valuable advice. Your opinion is very pertinent and we have benefited a lot. In the discussion part of the paper, we added a discussion on limitations and cited the paper. Straw burning is also one of the sources of agricultural carbon emissions. However, it is difficult to estimate the quantity of straw burning accurately. Due to data limitations, this paper does not consider the carbon emissions caused by open burning of crop residues when calculating the agricultural carbon emissions of various provinces in China. However, the agricultural carbon emission measurement system used in this paper fully considers the carbon emissions caused by animal breeding, rice planting and energy input in agricultural production, which is scientific and reasonable.

Reviewer 4 Report

The particular paper analyzes the internal relationship and 14 interaction mechanism among environmental governance, local government competition, and agri- 15 cultural carbon peak level. with the assistance of  a spatial panel model for  panel data from 30 provincial-level regions in the Chinese Mainland.

Interesting novel with significant scientific value a few amendments are necessary prior to acceptance for publication.

The manuscript is interesting novel that is not clearly phrased in the introduction section as well as in the abstract while also the scientific and socioeconomic value is not mentioned.

The methodlogy is a compositye one and it is not clear to me their implementation to the data, while in the results section limited comparison with the existing literature is provided.

A subje of future survey is missin in the last section along with study limitations

Author Response

To reviewer 4:

Dear reviewer and editor,

Thank you very much for your valuable advice. Your opinions are very important. We have carefully revised the following according to the review opinions of the paper.

Point 1: The manuscript is interesting novel that is not clearly phrased in the introduction section as well as in the abstract while also the scientific and socioeconomic value is not mentioned.

Thank you for your valuable advice. Your opinion is very pertinent and we have benefited a lot. According to your suggestion, we have added the scientific and socio-economic value of the paper in the abstract and introduction.

Point 2: The methodlogy is a compositye one and it is not clear to me their implementation to the data, while in the results section limited comparison with the existing literature is provided.

Thank you for your valuable advice. Our team has published the measurement method of agricultural carbon emissions in the top Chinese journals in the field of agricultural economy in the early stage, and the scientific nature of the method has been recognized by relevant experts in the industry. According to your suggestion, we have added the discussion part to the paper, made a comparative analysis between the paper and the existing research, and proposed the limitations of the paper and the direction for further research.

Point3: A subje of future survey is missin in the last section along with study limitations

Thank you for your valuable advice. According to your suggestion, we have added the discussion part to the paper, made a comparative analysis between the paper and the existing research, and proposed the limitations of the paper and the direction for further research.

Reviewer 5 Report

The paper has interesting proposals and results as well., but there is a lack of information about its database and basic information that should be presented.

The basic idea of a scientific paper is its repeatability, so the author should explain from where came ALL information and data.

The information in the introduction from line 1 to line 70 must be referenced – as an international journal all information should be explicated by references. Not just the general information about China but also de examples (lines 68-70).

Maybe brief information about what is “carbon peak (goal)” will help the general reader to better understand the paper construction.

Again, the information in lines 78-82 must be referenced…. And again the lines 85-90. I understood that reference 9 is just about the phrase lines 83-85.

Line 100 “some scholars” lead to just one reference: 20 Maybe another language construction will clarify that it refers to 21, 22, and 23 too.

But “other scholars” expression lead just to reference 24 (e.g. Zhembo is one scholar!)

Again lines 111-127 include data without references.

 Please explain to an international reader where is the Chines Mainland.

 3.2. Data Source …

All data sources MUST BE REFERENCED, i.e. websites and yearbooks must be listed in reference or in a final annexe.

What exactly is the "480 research sample" (line 275)

 Although the authors explain the variables l. 300 and l.308-301, I would suggest using again the variable name between parenthesis also in the discussion of the results. e.g l. 377 …. environmental governance (envpro) …

I suggest adding in the conclusion and section about academic contribution/implications, bringing again the position of this paper among other literature. 

Author Response

To reviewer 5:

Dear reviewer and editor,

Thank you very much for your valuable advice. Your opinions are very important. We have carefully revised the following according to the review opinions of the paper.

Point 1: The information in the introduction from line 1 to line 70 must be referenced – as an international journal all information should be explicated by references. Not just the general information about China but also de examples (lines 68-70).

Thank you for your valuable advice. According to your suggestion, we have revised this part of the paper.

Point 2: Maybe brief information about what is “carbon peak (goal)” will help the general reader to better understand the paper construction.

Thank you for your valuable advice. According to your suggestion, we have added the concept of carbon peak when explaining the agricultural carbon peak level.

Point 3: Again, the information in lines 78-82 must be referenced…. And again the lines 85-90. I understood that reference 9 is just about the phrase lines 83-85.

Line 100 “some scholars” lead to just one reference: 20 Maybe another language construction will clarify that it refers to 21, 22, and 23 too.

But “other scholars” expression lead just to reference 24 (e.g. Zhembo is one scholar!)

Again lines 111-127 include data without references.

Thank you for your valuable advice. According to your suggestion, we have revised this part of the paper.

Point 4:  Please explain to an international reader where is the Chines Mainland.

Thank you for your valuable advice. In this paper, mainland China refers to other regions excluding Hong Kong, Macao and Taiwan. According to your suggestion, we have explained the term.

Point 5: All data sources MUST BE REFERENCED, i.e. websites and yearbooks must be listed in reference or in a final annexe.

Thank you for your valuable advice. According to your suggestion, we have annotated the data sources mentioned in the text in the reference section.

Point 6: What exactly is the "480 research sample" (line 275)

Thank you for your valuable advice. We have explained in the text that the data of 30 provinces in 16 years constitute 480 observations。

 Point 7: Although the authors explain the variables l. 300 and l.308-301, I would suggest using again the variable name between parenthesis also in the discussion of the results. e.g l. 377 …. environmental governance (envpro) …

Thank you for your valuable advice. According to your suggestion, we have revised this part of the paper.

Point 8: I suggest adding in the conclusion and section about academic contribution/implications, bringing again the position of this paper among other literature. 

Thank you for your valuable advice. According to your suggestion, we have added the discussion part to the paper, made a comparative analysis between the paper and the existing research, and proposed the limitations of the paper and the direction for further research.

Round 2

Reviewer 3 Report

I believe the authors have clearly addressed my comments. I recommend the publication of the article.

Reviewer 4 Report

To be published as it is